# Exploring Metabolic Pathways of Anamorelin, a Selective Agonist of the Growth Hormone Secretagogue Receptor, via Molecular Networking

**DOI:** 10.3390/pharmaceutics15122700

**Published:** 2023-11-29

**Authors:** Young Beom Kwak, Jeong In Seo, Hye Hyun Yoo

**Affiliations:** 1Korea Racing Authority, Gwachon 13822, Republic of Korea; kwakyoungbeom1301@gmail.com; 2Institute of Pharmaceutical Science and Technology, College of Pharmacy, Hanyang University, Ansan 15588, Republic of Korea; seojeongin@hanyang.ac.kr

**Keywords:** anamorelin, metabolism, molecular networking, LC-MS/MS, GHSR agonist

## Abstract

In this study, we delineated the poorly characterized metabolism of anamorelin, a growth hormone secretagogue receptor agonist, in vitro using human liver microsomes (HLM), based on classical molecular networking (MN) and feature-based molecular networking (FBMN) from the Global Natural Products Social Molecular Networking platform. Following the in vitro HLM reaction, the MN analysis showed 11 neighboring nodes whose information propagated from the node corresponding to anamorelin. The FBMN analysis described the separation of six nodes that the MN analysis could not achieve. In addition, the similarity among neighboring nodes could be discerned via their respective metabolic pathways. Collectively, 18 metabolites (M1–M12) were successfully identified, suggesting that the metabolic pathways involved were demethylation, hydroxylation, dealkylation, desaturation, and *N*-oxidation, whereas 6 metabolites (M13a*-b*, M14a*-b*, and M15a*-b*) remained unidentified. Furthermore, the major metabolites detected in HLM, M1 and M7, were dissimilar from those observed in the CYP3A4 isozyme assay, which is recognized to be markedly inhibited by anamorelin. Specifically, M7, M8, and M9 were identified as the major metabolites in the CYP3A4 isozyme assay. Therefore, a thorough investigation of metabolism is imperative for future in vivo studies. These findings may offer prospective therapeutic opportunities for anamorelin.

## 1. Introduction

Anamorelin is a ghrelin receptor agonist that stimulates the growth hormone secretagogue receptor (GHSR), that has been developed for treating anorexia and weight loss. It is also known as ONO-7643, RC-1291, or ST-1291 [1,2,3,4,5]. Anamorelin has shown promising outcomes in clinical trials targeting the anorexia and weight loss associated with advanced pancreatic cancer. However, it failed to receive marketing approval in Phase 3 clinical trials [1,6]. Despite the previous setback, investigations on similar drugs are ongoing, and further research on anamorelin is exploring methods to surmount past limitations regarding its use as a treatment by appraising its pros, cons, and side effects. In this regard, revealing anamorelin’s metabolic mechanism in detail could offer insights into its advantages and disadvantages [7,8]. This is because studying metabolic mechanisms can facilitate the identification of and improvement in the therapeutic effects of drugs, thereby discovering and developing their potential applications in treating other diseases. However, at present, publicly reported data on the metabolism of anamorelin are unavailable.

Molecular network analysis is a computational strategy that can be used to visualize and interpret complex data resulting from MS analysis using the Global Natural Products Social Molecular Networking (GNPS) Platform (https://gnps.ucsd.edu/ProteoSAFe/static/gnps-splash.jsp (accessed on 23 November 2023)). Classical molecular networking (MN) can identify potential similarities between all MS/MS spectra in a dataset and propagate annotations to unknown but structurally related molecules [9]. More specifically, MN analysis utilizes the similarity of the fragmentation patterns of molecules to determine their correlation [10]. Thus, it could facilitate the workflow of drug metabolism research compared with traditional methods. Consequently, there has been a notable increase in the number of studies adopting the MN approach in recent times [9,11]. Feature-based molecular networking (FBMN) is an advanced version of MN. The advantage of FBMN over MN is its utilization of a well-established MS processing software for data pre-processing, allowing for the incorporation of not only MS information, such as isotope patterns, but also chromatographic characteristics, such as retention times, peak width, and resolution. This incorporation enables compounds with similar MS^2^ spectra but different retention times to be distinctively included in the network. Additionally, the relative intensity information of the spectrum is semi-quantified and displayed on the node, providing researchers with additional molecular information [12].

In the present study, our goal was to comprehensively delineate the largely unexplored metabolic profile of anamorelin to enhance its potential clinical success. To achieve this, we identified anamorelin metabolites using human liver microsomes (HLM) in vitro. Additionally, the structural information obtained was validated through additional experiments (the flavin-containing monooxygenases (FMO) incubation assay and utilization of an atmospheric pressure chemical ionization (APCI) source).

## 2. Materials and Methods

### 2.1. Materials

Anamorelin was provided by Toronto Research Chemical, Inc. (North York, ON, Canada). Pooled HLM, c-DNA-expressing CYP3A4, and three human flavin-containing monooxygenases (FMO) isoforms (FMO1, FMO3, and FMO5) were obtained from BD Gentest Corp. (Woburn, MA, USA). Glucose 6-phosphate (G6P), β-NADP^+^, glucose 6-phosphate dehydrogenase (G6Pd), and ammonium formate were obtained from Sigma Chemical Co. (St. Louis, MO, USA). All chemicals were analytical grade and were used as received. HPLC-grade acetonitrile (ACN) and distilled water (DW) were purchased from J.T. Baker (Phillipsburg, NJ, USA). Formic acid was purchased from Junsei Chemical Co. (Chou-ku, Japan). 

### 2.2. Microsomal and cDNA-Expressed Recombinant CYP Isozyme Incubation Assay

In vitro metabolites of anamorelin were investigated using 1 mg/mL of microsomal protein in 0.1 M potassium phosphate buffer (pH 7.4) at 37 °C for 2 h. The reaction was initiated by adding an NADPH-generating system (NGS) containing 10 mg/mL β-NADP^+^, 0.1 M G6P, and 1 unit/mL of G6Pd to the reaction mixture. After incubation, the reaction was terminated by the addition of ice-cold ACN. The mixture was kept on ice followed by centrifugation for 5 min at 13,200 rpm. An aliquot (10 μL) of the supernatant was used for ultra-high-performance liquid chromatography (UHPLC) Q-Orbitrap analysis. All experiments were performed in duplicate.

Additionally, the anamorelin metabolism pathway was investigated using a cDNA-expressed recombinant CYP3A4 isozyme. Incubation was performed under the same conditions described above in the presence of recombinant CYP3A4.

### 2.3. LC/MS/MS Analysis of Anamorelin

The UHPLC Q-Orbitrap system consisted of a Thermo Ultimate 3000 UHPLC (Thermo Fisher Scientific, Bremen, Germany) and a Q-Exactive^TM^ mass spectrometer (Thermo Fisher Scientific, Bremen, Germany). The column used for separation was a Phenomenex Luna Omega^TM^ 1.6 μm column (2.1 × 100 mm). The column temperature was maintained at 50 °C using a thermostatically controlled column oven. The mobile phase consisted of 5 mM ammonium formate (pH 3.0) in DW (Solvent A) and 0.1% formic acid in ACN (Solvent B). A gradient program was used for liquid chromatographic separation at a flow rate of 0.25 mL/min. For anamorelin and metabolite analysis, solvent B was 0% as the initial condition (t = 0 min), held for 5 min, and then increased to 70% from t = 5 min to t = 23 min; solvent B was returned to 0% from t = 23.0 min to t = 23.1 min, and stabilized until t = 25 min before the next injection. The column eluent was introduced directly into the mass spectrometer. UHPLC was coupled with the MS via a heated ESI source (HESI-II) operated in the positive ionization mode. For drug metabolite detection, the positive ESI mode was used. The ionization voltage was set at 3.5 kV. Mass calibration was performed, and the HESI-II source was used with heating (413 °C). The acquisition modes employed were each full scan and parallel reaction monitoring (PRM). The full scan acquisition range was set to *m*/*z* 150–800 Da for the positive mode, with a mass resolution of 70,000 (FWHM) and automatic gain control (AGC) of 3 × 10^6^. A mass resolution of 17,250 (FWHM) for the PRM mode was used with an isolation window of 0.8 amu. For MN analysis, the data-dependent MS^2^ (ddMS^2^) mode was applied to trigger the fragmentation with a normalized collision energy (NCE) of 40 eV. The 12 highest parent ions at each scan point of MS were selected as the target precursor ions for further MS^2^ fragmentation. The acquisition range was set to *m*/*z* 150–600 Da for the positive mode, with a mass resolution of 35,000 (FWHM) and automatic gain control (AGC) of 2 × 10^4^.

### 2.4. MN and FBMN Analyses for Metabolite Screening

For MN generation, the raw data files obtained from the ddMS^2^ analysis were converted from .raw to .mzXML format using MSConvert (http://proteowizard.sourceforge.net (accessed on 23 November 2023)). The .mzXML file was uploaded to the Global Natural Product Social Molecular Networking (GNPS) web-based platform through WinSCP (version 5.15.3) and analyzed using GNPS (http://gnps.ucsd.edu (accessed on 23 November 2023)). The FBMN was created according to the workflow of the GNPS platform [12]. The mzXML input file was pre-processed and converted into output files (.mgf and _quant.csv) that included the peak isolation sample information and strengths of MS and MS^2^ matched using the MZmine 2 (version MZmine 2.53) software. The output file was uploaded to the GNPS platform. For both MN and FBMN analyses in the GNPS platform, the base parameters were set to *m*/*z* 0.02 for the mass tolerance of the precursor and fragment ions. The minimum cluster size was set to 2. Links were also made between nodes when the cosine value was greater than 0.50 and the minimum number of common fragment ions matched by the MS/MS spectrum was 3. The MN and FBMN data were visualized and annotated using the Cytoscape 3.7.2 software (San Diego, CA, USA). Figure 1 outlines the sequential steps comprising the workflow for each molecular networking analysis.

### 2.5. FMO Incubation Assay

For the identification of the *N*-oxide metabolite, anamorelin was incubated with 0.5 mg/mL FMO isoforms (FMO1, FMO3, and FMO5) at 37 °C for 2 h. The reaction was initiated with NGS containing 10 mg/mL β-NADP^+^, 0.1 M G6P, and 1 unit/mL of G6Pd in the reaction mixture. After incubation, the reaction was terminated by the addition of ice-cold ACN. The mixture was kept on ice followed by centrifugation for 5 min at 13,200 rpm. An aliquot (10 μL) of the supernatant was used for UHPLC Q-Orbitrap analysis. All experiments were performed in duplicate.

### 2.6. APCI/MS Analysis

Atmospheric pressure chemical ionization (APCI) analysis was performed to confirm the presence of additional *N*-oxide metabolites. The instrument and column separation conditions were identical to those in the method described above. The operating conditions for positive mode APCI/MS were as follows: mass resolution (FWHM), 35,000; sheath gas flow rate, 31; auxiliary gas flow rate, 31; sweep gas flow rate, 0 (arbitrary units); discharge current, 4.97 µA; capillary temperature, 320 °C; S-lens RF level, 40; and automatic gain control, 5 × 10^6^.

## 3. Results

### 3.1. Anamorelin Metabolite Screening with Molecular Networking

To screen for anamorelin metabolites, we utilized MN analysis with ddMS^2^ data on the GNPS platform. Within the resulting networks, our initial search focused on identifying a network containing the parent drug’s node. This approach was based on the expectation that metabolites would likely share structural similarities and could thus be integrated into the same network as the parent drug [13]. Therefore, we identified the parent node with an *m*/*z* value of 547.3389 in a distinct network that was connected to 11 additional nodes ranging from 260.191 to 591.331 through our initial MN analysis of anamorelin metabolites (Figure 2A). However, during the manual inspection of the extracted ion chromatograms (EICs) for each node we observed several extra peaks eluted at different times, which could indicate the presence of isomers. To obtain a more accurate understanding of the metabolite landscape for anamorelin, we performed FBMN analysis, which enabled us to distinguish overlapping nodes. As a result, in the FBMN analysis, six nodes (563.336, 579.333, 549.318, 561.322, 575.339, 591.331) in the MN analysis were clearly separated into their respective nodes (Figure 2B).

### 3.2. Anamorelin Metabolite Structural Elucidation

The FBMN analysis yielded 18 metabolites (M1-M12) and 6 unknown compounds (M13a*-b*, M14a*-b*, and M15a*-b*), including anamorelin. To gain insight into the metabolic pathways of anamorelin, we examined the structures of each metabolite by analyzing their chromatographic characteristics and mass spectrum information, further comparing them to those of anamorelin (Appendix A, Figure 3). M1 (C_30_H_41_N_6_O_3_; 533.3235) exhibited a 14 amu decrease from the parent compound (anamorelin; C_31_H_43_N_6_O_3_; *m*/*z* 547.3391), indicating demethylation. M2 (C_30_H_39_N_6_O_3_; 531.3078), with a 16 amu decrease from the parent, suggested both demethylation and desaturation. The product ion of each metabolite, *m*/*z* 262 (C_15_H_24_N_3_O) for M1 and *m*/*z* 260 (C_15_H_22_N_3_O) for M2, respectively, was 14 and 16 amu smaller than that of the parent (*m*/*z* 276), signifying the loss of the methyl group (−14 amu) and di-hydrogen (−2 amu) from the di-methylamine group of anamorelin. M3a-b (C_31_H_43_N_6_O_4_; *m*/*z* 563.334) were attributed to the hydroxylation of the parent. The *m*/*z* 148 in the MS^2^ of M3a originated from the oxidation of the indole nitrogen. M4a-c (C_31_H_43_ N_6_O_5_; 579.3289) were metabolites with di-hydroxylation (+32 amu) of anamorelin. M5a (C_30_H_41_N_6_O_4_; *m*/*z* 549.3183) were demethylated metabolites (−14 amu) of M3a, in which demethylation occurred in the dimethylamine group, as indicated by the presence of *m*/*z* 262 in MS^2^. M5b-c (C_30_H_41_N_6_O_4_; *m*/*z* 549.3183) were demethylated metabolites (−14 amu) of M3b, in which demethylation occurred in the dimethylamine group, as indicated by the presence of *m*/*z* 262 in MS^2^. M6 (C_31_H_41_N_6_O_4_; *m*/*z* 561.3184) was determined as hydroxylation (+16 amu) and desaturation (−2 amu). M7 (C_16_H_26_N_3_O; *m*/*z* 276.207) was indicated as an *N*-dealkylated metabolite (−271 amu). Meanwhile, M8 (C_15_H_24_N_3_O, *m*/*z* 262.1914) and M9 (C_15_H_22_N_3_O, *m*/*z* 260.1757) exhibited *m*/*z* values that were 14 and 16 amu lower than M7, respectively. The metabolic patterns observed in M8 and M9 were similar to those of M1 and M2 from the parent compound, suggesting that demethylation, with and without desaturation steps at the dimethylamine group, occurred after *N*-dealkylation. Interestingly, we observed that M10 (C_31_H_43_N_6_O_4_; *m*/*z* 563.334) was eluted later than anamorelin on the chromatograms and 16 amu higher than the parent, which might indicate an *N*-oxidized metabolite. Additionally, M11a-b (C_31_H_43_N_6_O_5_; *m*/*z* 579.3289) exhibited *m*/*z* values that were 28 amu higher than the parent. M12 (C_30_H_41_N_6_O_4_; *m*/*z* 549.3183) exhibited *m*/*z* values that were 2 amu higher than the parent. M13a*-b* (C_32_H_43_N_6_O_4_; *m*/*z* 575.3345) and M14a*-b* (C_32_H_43_N_6_O_5_; *m*/*z* 591.3274) displayed increased CO and CO_2_ elemental compositions compared to the parent. M15a*-b* exhibited demethylation, accompanied by the addition of CO (Figure 4). But the interpretation of this observation was unclear. Methylation is not expected to occur during phase I biotransformation with NADPH fortification. The methylation of nitrogen-containing compounds in liver microsomes has been reported in several studies, but its applicability to this study is limited [14,15]. The asterisks used in compounds M13a*-b*, M14a*-b*, and M15a*-b* indicate substances whose biotransformation status under the in vitro microsomal system of anamorelin was unclear. Finally, the structurally characterized metabolites (M1–M12) were semi-quantitatively compared by measuring the area on the chromatogram, and as a result, M1 and M7 were found to be the major metabolites in HLM in vitro (Figure 5).

### 3.3. Confirmation of N-Oxide Metabolites and Identification of CYP3A4-Mediated Metabolites

Our structural analysis led us to predict M10 as an *N*-oxide metabolite due to the later retention time (Figure 5) and higher *m*/*z* value (+16 amu) compared to the parent. Additionally, we suspected that M11a-b and M12 were metabolites generated through *N*-oxidation-mediated metabolism. However, the complexity of the metabolic process in HLM made it challenging to confidently determine the presence of *N*-oxidized metabolites. Therefore, we conducted an additional assay to verify the formation of *N*-oxidated metabolites using FMO [16,17]. As a result of the FMO incubation assay using three isozymes (FMO1, FMO3, and FMO5), M10 was produced in all FMOs tested (Figure 6A). To further investigate the multistep metabolism mediated by both FMO and CYP, we modified our LC-MS system by replacing the ESI source with an APCI source. By utilizing an APCI source, we were able to identify metabolites derived from *N*-oxidation as an intermediate reaction via thermal deoxygenation during thermal energy activation in the vaporizer (Figure 6B) [18]. This method confirmed that M11a-b and M12 were the hydroxylated and demethylated metabolites generated after *N*-oxidation by FMO.

A recent study reported that anamorelin is metabolized by CYP3A4, predominantly present in the liver. Furthermore, CYP3A4 inhibitors have been reported to increase the AUC of anamorelin. However, this study does not provide detailed information regarding the metabolic process of anamorelin [19,20,21]. To compare the metabolite profiles observed in the HLM metabolism study, we conducted additional experiments using the CYP3A4 isozyme. Our findings revealed that CYP3A4 primarily produced *N*-alkylated metabolites (Figure 7), specifically M7, M8, and M9, despite M1 and M7 being identified as the major metabolites in HLM. 

## 4. Discussion

Drug metabolism profiling has traditionally been a challenging task due to the numerous possible metabolic pathways and the vast datasets that require manual inspection, making it time-consuming and expensive. This increases the likelihood that critical metabolites undergoing unusual pathways may be overlooked, leading to misinterpretations during the early preclinical stage. To address these challenges, molecular networking has emerged as a promising tool by providing a network of structurally relevant molecules in a semi-automatic fashion.

Initially, we employed MN analysis to quickly obtain snapshots of the novel anamorelin metabolites, as it has easy accessibility and a simple workflow that does not require data pre-processing. The MN analysis enabled us to identify 11 anamorelin metabolites (Figure 2). However, during our validation process, by checking the profile on the chromatograms, we observed multiple peaks in the same EIC, indicating the presence of overlapping nodes due to the isomers. This prompted us to use FBMN to unravel the metabolites of anamorelin in a more reliable manner. FBMN has several advantages over MN, including its ability to distinguish compounds that generate similar MS^2^ spectra by incorporating their chromatographic separation, enhancing spectral annotation, and providing semi-quantitative information that aids the metabolomic statistical evaluation by reflecting peak abundance [12]. Through FBMN analysis, we were able to separate six nodes (563.336→M3a-b, M10; 579.333→M4a-c, M11a-b; 549.318→M5a-c; M12; 575.339→M13a*-b*; 591.331→M14a*-b*; 561.322→M6, M15a*-b*) that could not be achieved using MN analysis (Figure 2). 

Based on the findings obtained from the FBMN analysis and by investigating fragmentation patterns, it was suggested that anamorelin undergoes hydroxylation, demethylation, and dealkylation in HLM (Figure 2B). Furthermore, the distinct chromatographic behavior of M10, M11a-b, and M12, in contrast to M3a-b, M4a-c, and M5a-c, respectively, suggested the involvement of additional metabolic mechanisms in their formation (Figure 5). Subsequent FMO incubation assays and the utilization of the APCI source showed that the *N*-oxidation through FMO was responsible for the generation of the terminal (M10) or *N*-oxidation-mediated metabolites (M11a-b and M12), as depicted in Figure 6. Overall, these findings allowed for the complete elucidation of the metabolic pathways of anamorelin in HLM, as summarized in Figure 8. 

Comprehending metabolic processes at the molecular level is paramount for the success of drug discovery and development [22]. This understanding plays a pivotal role in enhancing the stability, in vivo half-life, and the risk–benefit ratio of potential drug candidates [23]. Moreover, to mitigate the risk of expensive setbacks in the clinical stages stemming from insufficiently characterized drug metabolism, a thorough investigation of metabolic profiles is of utmost importance. This is because drug metabolism can lead to the generation of metabolites with markedly distinct physicochemical and pharmacological attributes compared to the parent drug, and these metabolites are associated with the drug’s effectiveness and safety [24,25]. Hence, our in-depth insights into the metabolites of anamorelin would be valuable for reevaluating and reformulating it, with the aim of investigating new therapeutic applications. Additionally, the present method, employing molecular networking for metabolite identification, has the potential for broader adoption in other drug metabolism studies. This method offers the benefit of accelerating the process while providing a near-complete coverage of metabolite formation.

Anamorelin has been conventionally recognized to predominantly undergo hepatic metabolism mediated by CYP3A4 [19,20]. In a recent study, experimental validation was conducted in healthy human volunteers, demonstrating that the concomitant administration of a CYP3A4 inhibitor or inducer can result in a marked alteration of anamorelin’s pharmacokinetic properties (i.e., maximum plasma concentration (C_max_) and area under the plasma concentration–time curve (AUC_0–∞_)) [21]. However, our investigation uncovered a discrepancy in the metabolite profiles of anamorelin between HLM and CYP3A4 isozymes. Specifically, we observed that only *N*-alkylation-mediated metabolites (M7, M8, and M9) were detectable in the CYP3A4 isozyme assay (Figure 7). This contrasts with the results obtained from HLM, which indicated that demethylation (M1) could be a primary metabolic pathway, along with *N*-dealkylation (M7), while other metabolites were still exhibiting significant peak intensities (Figure 5). These findings are in line with a prior report suggesting that demethylation plays a significant role in the metabolism of anamorelin [26]. Furthermore, given the diverse nature of CYP enzymes [27], it is plausible to hypothesize that other CYP isozymes may be involved in anamorelin’s metabolism. This raises important concerns regarding potential CYP-mediated drug–drug interactions, which could contribute to the reported adverse event of anamorelin, cardiac toxicity, due to its narrow therapeutic window characterized by non-linear pharmacokinetic profiles [26]. To mitigate potential adverse events from uncertainties, it is advisable not to rely solely on CYP3A4 as the target enzyme for studying anamorelin’s metabolism, as this approach may have limitations. Future investigations should be conducted to comprehensively explore other CYP isozymes that might be closely associated with anamorelin’s metabolism. Consequently, it is prudent to consider various metabolic pathways to achieve a thorough understanding of the therapeutic efficacy and safety of anamorelin.

## 5. Conclusions

In this study, we conducted a comprehensive investigation of the in vitro metabolism of anamorelin using HLM and analyzed the results using MN and FBMN approaches. Through these methods, we identified 18 metabolites (M1–M12) and 6 unknown compounds (M13a*-b*, M14a*-b*, and M15a*-b*) that were produced from anamorelin. Our analysis revealed that M1 and M7 were the primary metabolites, and we classified the metabolic pathways of anamorelin in HLM as hydroxylation, demethylation, and dealkylation. We also used the FMO incubation assay and thermal deoxygenation in the APCI source to investigate the complex nature of FMO-mediated metabolism. Our results indicated that hydroxylation and demethylation may occur after *N*-oxidation. Additionally, our CYP3A4 isozyme assay suggested that M7, M8, and M9 may also be the major forms of anamorelin metabolites in vivo, in addition to M1. Overall, these findings provide new insights into the metabolism of anamorelin and its potential as a novel therapeutic agent.

## Figures and Tables

**Figure 1 pharmaceutics-15-02700-f001:**
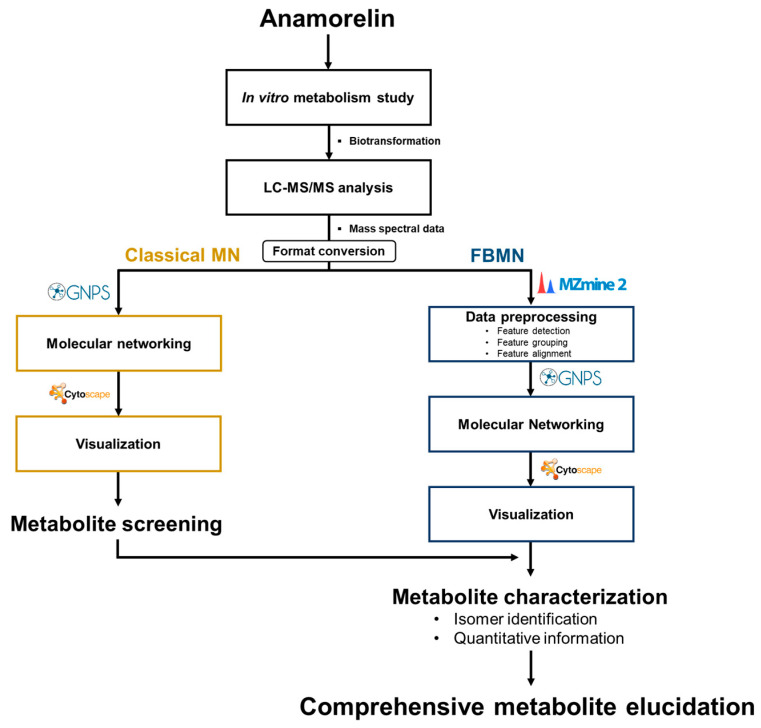
Molecular networking analysis flow chart.

**Figure 2 pharmaceutics-15-02700-f002:**
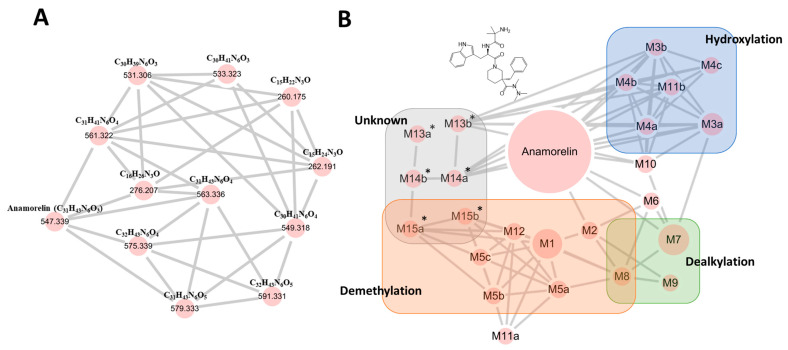
Molecular networking maps. (**A**) MN and (**B**) FBMN results of anamorelin and its metabolites generated from human liver microsome reaction for 2 h. The molecular networking result can be visualized directly on GNPS via the following links: https://gnps.ucsd.edu/ProteoSAFe/result.jsp?view=network_displayer&componentindex=15&highlight_node=1281&task=53b690a71dec466aa9a777916e8c0113#%7B%7D and https://gnps.ucsd.edu/ProteoSAFe/result.jsp?view=network_displayer&highlight_node=60096&componentindex=1&task=7fabfeb68f9c4fb2bb3e36bbfe8133a1#%7B%7D, accessed on 23 November 2023.

**Figure 3 pharmaceutics-15-02700-f003:**
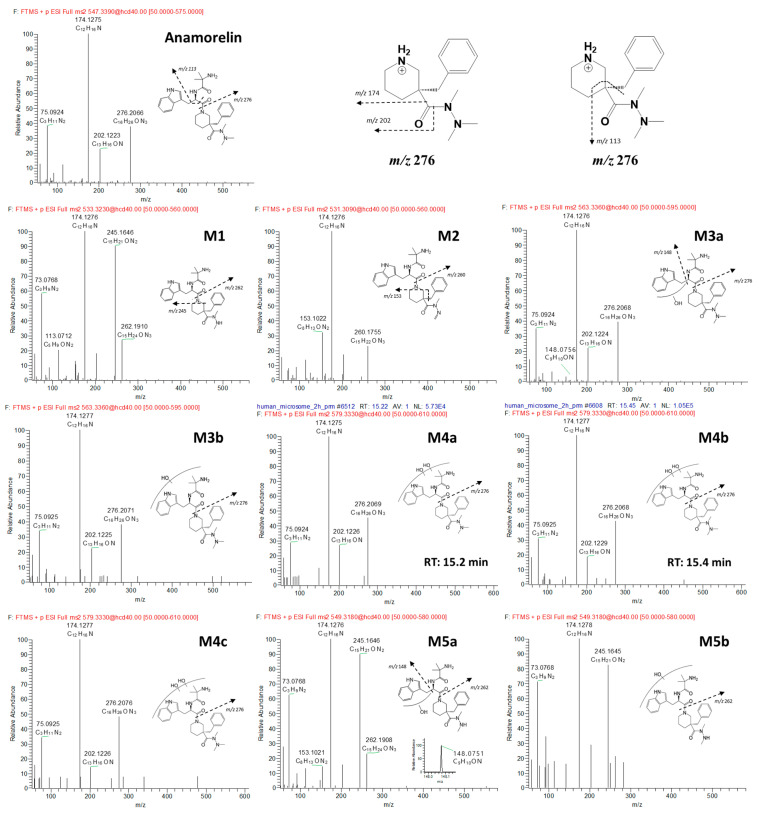
Representative MS/MS spectra and the proposed structure of each anamorelin metabolite generated from human liver microsome reaction for 2 h.

**Figure 4 pharmaceutics-15-02700-f004:**
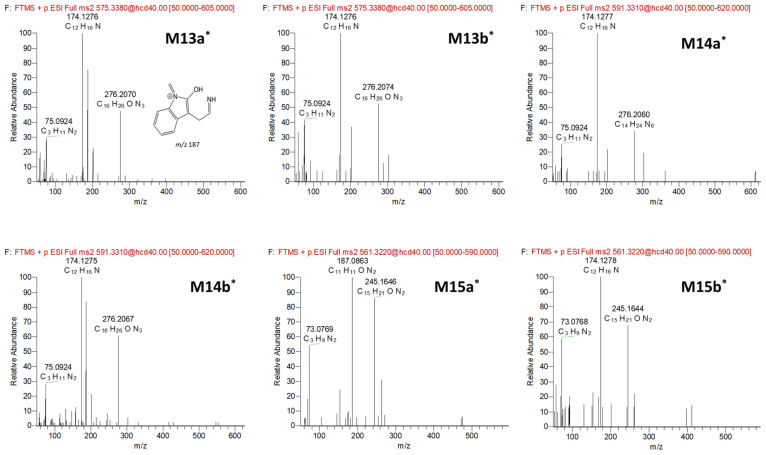
MS/MS spectra and the predicted structure of each unknown compound generated from human liver microsome reaction for 2 h.

**Figure 5 pharmaceutics-15-02700-f005:**
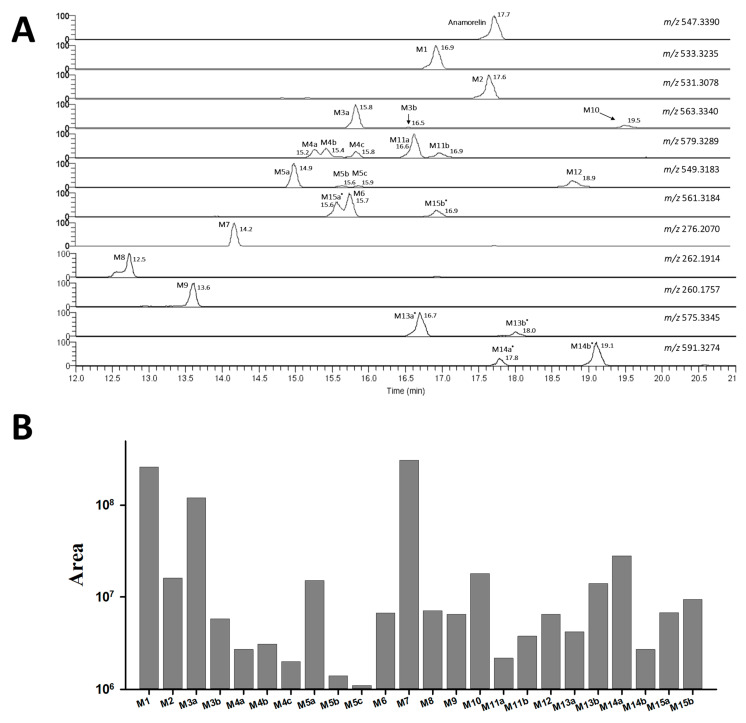
Chromatographic profiles of anamorelin metabolites. (**A**) Extracted ion chromatograms and (**B**) peak areas of anamorelin and its metabolites generated from human liver microsome reaction for 2 h.

**Figure 6 pharmaceutics-15-02700-f006:**
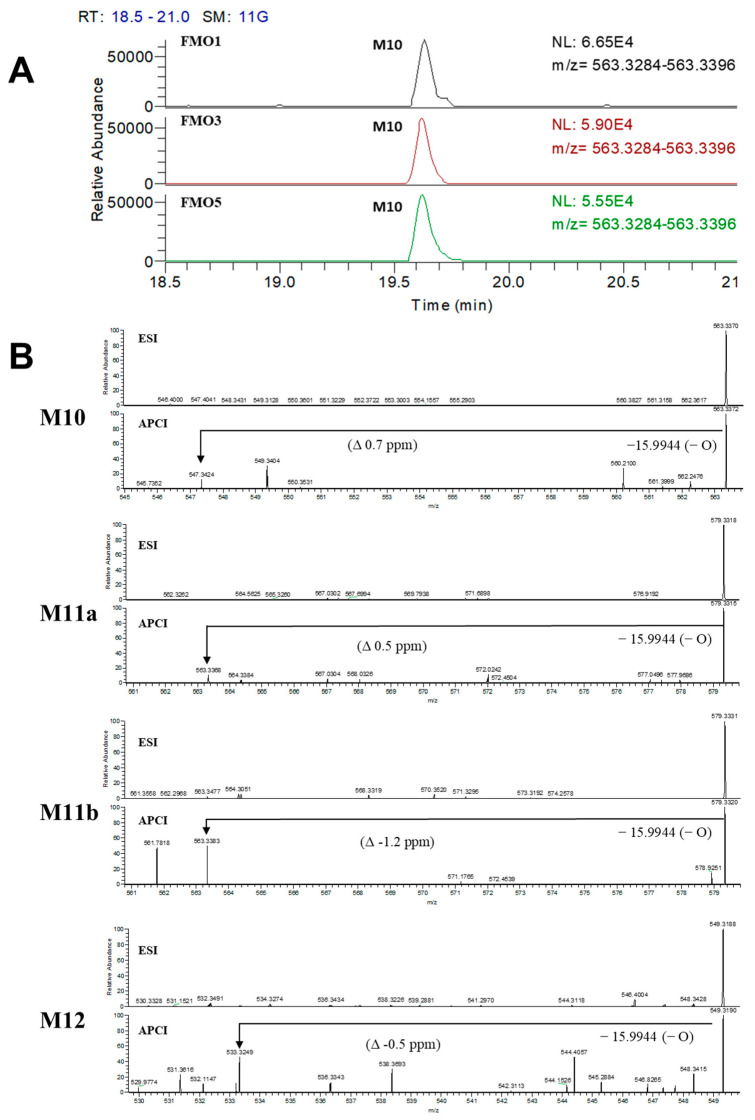
(**A**) M10 formation in the FMO incubation assay. (**B**) MS/MS spectra of M10, M11a, M11b, and M12 in ESI (top) and the thermal deoxygenation observed in the APCI (bottom) MS spectra.

**Figure 7 pharmaceutics-15-02700-f007:**
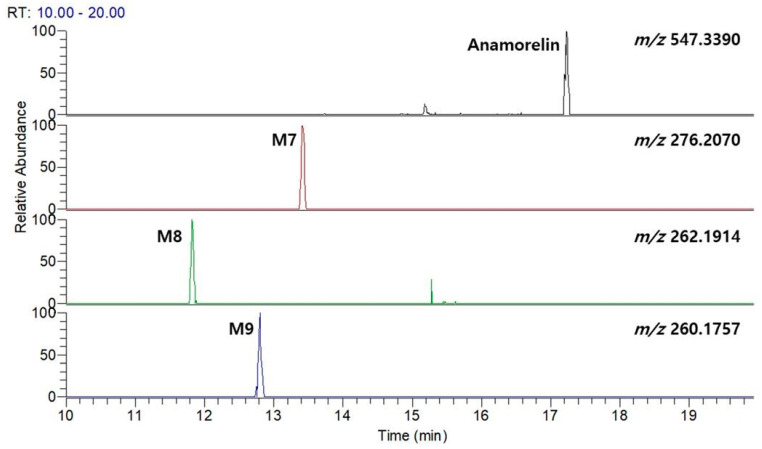
Extracted ion chromatograms of anamorelin and its metabolites generated from the incubation with cDNA-expressed recombinant CYP3A4 isozyme for 2 h.

**Figure 8 pharmaceutics-15-02700-f008:**
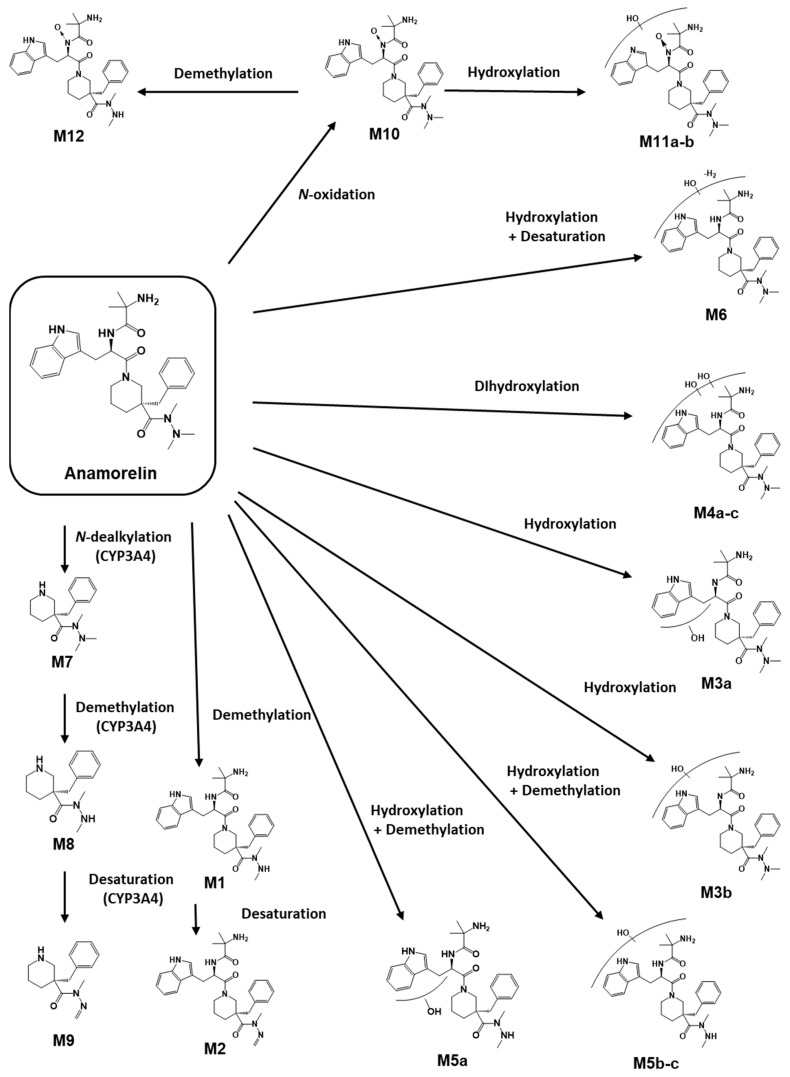
Postulated metabolic pathways of anamorelin in human liver microsomes.

## Data Availability

Data are contained within the article and Appendix A.

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
