# Peer review of "Exploring Metabolic Pathways of Anamorelin, a Selective Agonist of the Growth Hormone Secretagogue Receptor, via Molecular Networking"

_pharmaceutics, 2023, doi:10.3390/pharmaceutics15122700_

Round 1

Reviewer 1 Report

Comments and Suggestions for Authors

The manuscript titled "Investigating Anamorelin Metabolism via Molecular Networking: Unveiling Complex Pathways" provides a comprehensive exploration of the metabolic pathways of anamorelin, a selective growth hormone secretagogue receptor agonist. This investigation was carried out using human liver microsomes (HLM) and molecular networking (MN) techniques. While the study offers valuable insights into the compound's metabolism, several points need to be addressed and clarified for the manuscript to be suitable for publication.

 1. The introduction effectively establishes the context and importance of studying anamorelin's metabolism. However, it would be advantageous to explicitly state the specific research questions or objectives addressed in this study. Clarifying the knowledge gaps or research goals that the authors aimed to achieve will enhance the manuscript's focus.

2. Additional information regarding the methodology for conducting "Molecular Networking" is required. A more detailed description of the Molecular Networking process will aid readers in comprehending the steps involved in this analytical approach. This enhanced explanation will be valuable for researchers interested in replicating or applying similar methods in their own metabolomics studies.

3. The discussion of the significance and implications of the findings should be more comprehensive. It is important to explain the relevance of identifying these metabolites and pathways, emphasizing how they contribute to our understanding of anamorelin's pharmacology and potential clinical applications. Additionally, discuss the potential impact of this information on future drug development or the clinical use of anamorelin.

 4. The manuscript highlights differences in the metabolite profiles observed between HLM and CYP3A4 isozymes. This intriguing observation should receive a more in-depth discussion. Address whether these differences align with existing literature or if they raise questions about the compound's metabolic fate in vivo. Providing insights into the significance of these distinctions will enhance the manuscript's value.

Comments on the Quality of English Language

The manuscript's language and style need some improvement. There are grammatical and typographical errors throughout the text that should be corrected. Ensure consistent formatting and citation style.

Author Response

We thank the editor and the referees for their time and commitment to review our contribution. Please find the response to the referee’s points and suggestions below. Our responses are in blue and preceded with an →.

Reviewer #1

The manuscript titled "Investigating Anamorelin Metabolism via Molecular Networking: Unveiling Complex Pathways" provides a comprehensive exploration of the metabolic pathways of anamorelin, a selective growth hormone secretagogue receptor agonist. This investigation was carried out using human liver microsomes (HLM) and molecular networking (MN) techniques. While the study offers valuable insights into the compound's metabolism, several points need to be addressed and clarified for the manuscript to be suitable for publication.

  1. The introduction effectively establishes the context and importance of studying anamorelin's metabolism. However, it would be advantageous to explicitly state the specific research questions or objectives addressed in this study. Clarifying the knowledge gaps or research goals that the authors aimed to achieve will enhance the manuscript's focus.

→ We appreciate the reviewer’s comment. We have added a sentence explicitly stating our goal (Page 2; Line 60-62).

  1. Additional information regarding the methodology for conducting "Molecular Networking" is required. A more detailed description of the Molecular Networking process will aid readers in comprehending the steps involved in this analytical approach. This enhanced explanation will be valuable for researchers interested in replicating or applying similar methods in their own metabolomics studies.

→ We express our gratitude to the reviewer for bringing up this issue. In response to this feedback, we have added a schematic workflow of the methodology using molecular networking as Figure 1 in the Materials and Methods section along with the revised graphical abstract more clearly describing the present study. We believe that these visual representations will significantly improve the reader's grasp of our methodology and its potential applicability to their own research endeavors. Additionally, we previously stated the workflow of molecular networking (MN) and feature-based molecular networking (FBMN) in section 2.4. Also, the previously embedded links in the legend of Figure 2 (originally Figure 1), grant readers free access to information regarding our molecular networking analysis, including specific parameter settings on the GNPS website. As figure 1 has been added, the figure order was rearranged.

  1. The discussion of the significance and implications of the findings should be more comprehensive. It is important to explain the relevance of identifying these metabolites and pathways, emphasizing how they contribute to our understanding of anamorelin's pharmacology and potential clinical applications. Additionally, discuss the potential impact of this information on future drug development or the clinical use of anamorelin.

→ We appreciate the reviewer’s comment. We have added the paragraph stating the importance and implication of our findings mainly focusing on comprehensive anamorelin metabolite identification and the present methodology using molecular networking (Page 14; Line 298-311).

  1. The manuscript highlights differences in the metabolite profiles observed between HLM and CYP3A4 isozymes. This intriguing observation should receive a more in-depth discussion. Address whether these differences align with existing literature or if they raise questions about the compound's metabolic fate in vivo. Providing insights into the significance of these distinctions will enhance the manuscript's value.

→ We thank the reviewer for the comment. We have added more discussion regarding the discrepancy in metabolic profiles between HLM and CYP3A4 isozyme (Page 16; Line 322-337).

Reviewer 2 Report

Comments and Suggestions for Authors

The text „Exploring Metabolic Pathways of Anamorelin, a Selective Agonist of the Growth Hormone Secretagogue Receptor, via Molecular Networking” presents a systematic study of Anamorelin metabolites using biotransformation and LC-MS/MS analysis.

The design of experiments is interesting, the obtained results could be useful for further studies.

There are some questions that should be answered before the paper is further considered for publication:

1.     Please provide location for GNPS on first mention (line 44)

2.     Please specify other factors from chromatographic separation that were used, apart from retention time (see line 55)

3.     For MS/MS, both MS2 and MS2 are used, please check this for consistent use. Is the use of MS1 instead of MS necessary?

4.     The sentence (line 61) “Metabolites screened from the MN and FBMN analysis were presented with structures through MS1 and MS2 obtained from chromatography and ESI” shows strange order of data and analysis, please reconsider.

5.     Please clarify: line 95: ACN in 0.1 % formic acid (or 0.1% formic acid in ACN?)

6.     Please clarify the first stage of fragmentation (line 109: trigger the second stage of fragmentation) or rephrase this part.

7.     The references in part 2.4 should be consistent (line 121)

8.     The sentence (line 198-200) needs clarification, as EIC is selected for specific m/z, and the indicated compounds are  of different m/z values.

9.     Please explain the (*) in line 202

10.  The quantitative comparison (lines 207-209) does not take into account possible differences in ionizability. Please comment.

11.  The order of figures and table 1 is confusing, please consider starting from EIC. (4A, 2, table)

12.  Most data in table 1 is repeated from figure 2, the new information is retention time (then added in figure 4A) and transformation (figure 7). Table 1 takes a lot of space, please consider moving it to supplementary data.

13.  Figure 2, structure for m/z 202: the aldehyde form is not acceptable. Structure 174: location of unsaturated bond. How is m/z 113 formed?

14.  M2 – the proposed structure for m/z 153 is strange, did the Authors considered rearrangement?

15.  According to figure 2, there is no difference between M4a and M4b. retention time information is needed at this stage.

16.  Did the Authors considered oxidation of indole nitrogen? Is there any proof for oxidation of the indicated nirogen?

17.  Formation of M14a* and similar products – addition of CO2 versus CO and oxidation – did the Authors considered this possibility?

18.  Is there any reference on possible tryptophane metabolite similar to 13a/14a

19.  Figure 4A: in EIC 561, there are additional peaks, not marked. Please indicate the reason.

20.  Please compare the sentence in lines 40-41 and the statement in line 259. A more detailed description would be helpful.

21.  In fig. 7, the transformation M2 – M9 is not dealkylation, it is acyl removal.

Comments on the Quality of English Language

Line 35: please clarify: anamorelin is exploring methods to surmount its past limitations

Line 205: the end of sentence is missing? “during phase I biotransformation with NADPH-fortified.”

Author Response

Reviewer #2

The text „Exploring Metabolic Pathways of Anamorelin, a Selective Agonist of the Growth Hormone Secretagogue Receptor, via Molecular Networking” presents a systematic study of Anamorelin metabolites using biotransformation and LC-MS/MS analysis.

The design of experiments is interesting, the obtained results could be useful for further studies.

There are some questions that should be answered before the paper is further considered for publication:

  1. Please provide location for GNPS on first mention (line 44)

→ We have provided the location for GNPS on first mention. (L44-45)

  1. Please specify other factors from chromatographic separation that were used, apart from retention time (see line 55)

→ We have specified other factors in chromatographic separation and added them to the manuscript (L55).

  1. For MS/MS, both MS2 and MS2are used, please check this for consistent use. Is the use of MS1 instead of MS necessary?

→ MS1 was modified to MS. And MS2 was changed as MS2. The corrected content has been reflected in the manuscript.

  1. The sentence (line 61) “Metabolites screened from the MN and FBMN analysis were presented with structures through MS1 and MS2 obtained from chromatography and ESI” shows strange order of data and analysis, please reconsider.

→ This content has been deleted because it may cause confusion.

  1. Please clarify: line 95: ACN in 0.1 % formic acid (or 0.1% formic acid in ACN?)

→ The correct expression is 0.1% formic acid in ACN. We have corrected the typo. (L95-96)

  1. Please clarify the first stage of fragmentation (line 109: trigger the second stage of fragmentation) or rephrase this part.

→ The expression has been reorganized as follows. : “~ trigger the fragmentation with a normalized collision energy (NCE) of 40 eV. (L108-110)

  1. The references in part 2.4 should be consistent (line 121)

→ The reference format has been changed. (L120)

  1. The sentence (line 198-200) needs clarification, as EIC is selected for specific m/z, and the indicated compounds are of different m/z values.

→ M4a-c (C31H43N6O5; m/z 579.3289) and M11a-b (C31H43N6O5; m/z 579.3289) have the same element composition. M5a-c (C30H41N6O4; m/z 549.3183) and M12 (C30H41N6O4; m/z 549.3183) also have the same element compositions. The sentence has been revised to minimize confusion. (L203-205)

  1. Please explain the (*) in line 202

→ The asterisks used indicate compounds M13a*-b* and M14a*-b*, substances whose biotransformation status under the in vitro microsomal system of anamorelin is unclear.

  1. The quantitative comparison (lines 207-209) does not take into account possible differences in ionizability. Please comment.

→ We agree with your perspective. In the absence of standard substances, quantitative comparison of metabolites is not feasible. Therefore, we conducted a semi-quantitative comparison by measuring chromatographic peak areas. To avoid ambiguity, we will clarify the statement. (L214)

  1. The order of figures and table 1 is confusing, please consider starting from EIC. (4A, 2, table)

→ The order of figures and tables has been rearranged as recommended. The changes are as follows:

Figure 4 →Figure 3; Figure 2 → Figure 4; Table 1→ Table S1

  1. Most data in table 1 is repeated from figure 2, the new information is retention time (then added in figure 4A) and transformation (figure 7). Table 1 takes a lot of space, please consider moving it to supplementary data.

→ Table 1 has been moved to supplementary data (Table S1), and retention time information have been listed in Figure 3A.

-

  1. Figure 2, structure for m/z 202: the aldehyde form is not acceptable. Structure 174: location of unsaturated bond. How is m/z 113 formed?

→ We have relocated the predicted cleavage positions for m/z 202, 174 and 113 to the ionized structure at m/z 276 (Figure 3A).

  1. M2 – the proposed structure for m/z 153 is strange, did the Authors considered rearrangement?

→ We have checked the structure for m/z 153 but found it unclear, so we have deleted the structure of m/z 153

  1. According to figure 2, there is no difference between M4a and M4b. retention time information is needed at this stage.

→ We have indicated the retention time information for both M4a and M4b in Figure 4.

  1. Did the Authors considered oxidation of indole nitrogen? Is there any proof for oxidation of the indicated nirogen?

→ We considered the possibility of oxidation of the indole nitrogen. According to our speculation, if oxidation of the indole nitrogen had occurred, a fragment ion at m/z 148 should have been present. However, the m/z 148 fragment ion was not detected. Additionally, oxidation is known to more readily occur at secondary nitrogens rather than primary nitrogens. Therefore, based on this information, we have inferred the current position as the likely site of oxidation

  1. Formation of M14a* and similar products – addition of CO2 versus CO and oxidation – did the Authors considered this possibility?

→ In consideration of the reviewer's opinion, the expression CO2 has been changed to “CO + oxidation”. (Table S1)

  1. Is there any reference on possible tryptophane metabolite similar to 13a/14a

→ One of the prominent metabolic pathways for tryptophan is the kynurenine pathway, and various other metabolic processes, including 3-Indolepyruvic acid, Indole acetamide, Serotonin etc, have been known (10.1038/s41573-019-0016-5, 10.4137/IJTR.S12626, Tryptophan metabolism in man. The Journal of Clinical Investigation). The methylation reactions are anticipated based on the predicted elemental compositions of the generated products 13a and 14a. However, no discernible evidence could be found for the metabolic pathways and processes of tryptophan.

M13a*-b* (C32H43N6O4) shows an elemental composition with the addition of CO compared to the parent drug (C31H43N6O3). Similarly, M14a*-b* (C32H43N6O5) exhibits an elemental composition with the addition of CO and oxidation in comparison to the parent drug (C31H43N6O3). In general, methylation is not typically expected to occur during phase I biotransformation with liver microsomes. In this context, several reports suggest that methylated compounds of nitrogen-containing substances could be artifacts resulting from reactions with reagents used in in vitro microsome analysis (references: 10.1002/rcm.5005, 10.1124/dmd.105.008367).

  1. Figure 4A: in EIC 561, there are additional peaks, not marked. Please indicate the reason.

→ Additional peaks have been indicated (M15a* at 15.6 min; M15b* at 15.9). These peaks had similar fragmentation patterns to M13a*-b* and M14a*-b*. Thus, M15a*-b* exhibited a fragmentation pattern indicating demethylation and the addition of CO. This information has been incorporated into the revision manuscript (Figure 3, Figure 5, L207-208).

  1. Please compare the sentence in lines 40-41 and the statement in line 259. A more detailed description would be helpful.

→ The sentence has been specified as follows. Recent study has reported that anamorelin is metabolized by CYP3A4, predominantly present in the liver. Furthermore, CYP3A4 inhibitors have been reported to increase the AUC of anamorelin. However, the paper does not provide information about the details regarding the metabolic process (L255-258).

  1. In fig. 7, the transformation M2 – M9 is not dealkylation, it is acyl removal.

→ In this study, M7 was the main metabolite. Since the metabolic pathway from M2 to M9 is unclear, the entire pathway has been modified. (Figure 8)

Round 2

Reviewer 1 Report

Comments and Suggestions for Authors

I am generally satisfied with the author's response and have no further questions.